# Multiple risk factor control, mortality and cardiovascular events in type 2 diabetes and chronic kidney disease: a population-based cohort study

Shota Hamada,[1,2] Martin C Gulliford[1,3]

[1]Department of Primary Care and Public Health Sciences, King's College London, London, UK
[2]Research Department, Institute for Health Economics and Policy, Association for Health Economics Research and Social Insurance and Welfare, Tokyo, Japan
[3]National Institute for Health Research Biomedical Research Centre, Guy's and St Thomas' National Health Service Foundation Trust, London, UK

**Correspondence to**
Dr Shota Hamada;
shota.hamada@ihep.jp

## ABSTRACT

**Objectives** This study aimed to evaluate the effectiveness of multiple risk factor control (MRFC) at reducing mortality and cardiovascular events in diabetes and chronic kidney disease (CKD) in clinical practice.

**Design** Population-based cohort study.

**Setting** Primary care database in the UK, linked with inpatient and mortality data.

**Participants** Participants aged 40–79 years with type 2 diabetes and valid serum creatinine measurements, including 11 431 participants with CKD (estimated glomerular filtration rate: eGFR 15–59 mL/min/1.73 m$^2$) and 36 429 participants with non-CKD (eGFR ≥60 mL/min/1.73 m$^2$).

**Exposures** MRFC consisted of four components: Haemoglobin A1c (HbA1c) <53 mmol/mol (<7.0%), blood pressure <140/90 mm Hg, total cholesterol <5 mmol/L and no smoking. The main exposure variable was the number of risk factors controlled at baseline.

**Outcome measures** All-cause and cardiovascular mortality in the overall participants. Cardiovascular events, including coronary heart disease and stroke, in participants limited to those without a history of cardiovascular diseases at baseline.

**Results** In participants with CKD, 37% or 13% met three or four MRFC criteria, respectively. Increasing numbers of risk factors controlled were associated with lower relative hazards for all outcomes studied compared with those meeting no or one criterion. For participants with CKD meeting four criteria, the adjusted HR for all-cause mortality was 0.60 (95% CI 0.53 to 0.69) and the adjusted subdistribution HR for cardiovascular mortality was 0.60 (95% CI 0.50 to 0.70), considering a competing risk of non-cardiovascular death. Participants meeting four criteria also had lower relative hazards for coronary heart disease (adjusted subdistribution HR 0.73, 95% CI 0.59 to 0.91) and stroke (0.63, 95% CI 0.45 to 0.89), considering death as a competing risk.

**Conclusions** MRFC may lower the increased risks for mortality and cardiovascular events in people with diabetes and CKD. Further research is needed to evaluate appropriateness of MRFC according to individual participants' health status for improved management of cardiovascular risks in this population.

## INTRODUCTION

Diabetes and chronic kidney disease (CKD) are growing health problems worldwide,

### Strengths and limitations of this study

► This study included a large number of participants with type 2 diabetes and chronic kidney disease sampled from a representative general population with about 6 years of follow-up, which enabled to determine the associations of cardiovascular risk factors with mortality and cardiovascular events.
► Linked data for diagnostic data in hospitals and death registration with a primary care database enhanced the validity of the study to evaluate mortality and cardiovascular events.
► We could not conclude that associations represented causal relationships between multiple risk factor control and mortality and cardiovascular events in this non-randomised study.
► There is a possibility of confounding if healthier participants were managed more successfully and this resulted in being categorised as those with greater number of risk factors controlled.

contributing to increased mortality.[1] Diabetes and CKD also impose a substantial economic burden on society, with particularly high costs relating to cardiovascular complications and renal replacement therapy.[2 3] The prevalence of CKD in patients with diabetes is between 4.2% and 17.9% (CKD stages 3–5) in European countries.[4] The leading cause of death in people with type 2 diabetes or CKD is cardiovascular disease rather than renal complications.[5 6] Prevention of cardiovascular events is a key focus in the management of patients with these conditions.

Multifactorial interventions to reduce cardiovascular risks were shown to be effective at reducing mortality and cardiovascular events in patients with type 2 diabetes and persistent microalbuminuria in the Steno-2 randomised trial.[7 8] This study provided a high level of evidence, but included a relatively small number of participants with diabetes who were managed in specialist centres. Recently, the implementation and

effectiveness of this approach have been evaluated in patients with diabetes in clinical practice settings.[9–11] Epidemiological studies have demonstrated additional risks of CKD on mortality and cardiovascular diseases in people with diabetes,[12] but treatment approaches in this population have not been well studied. No studies focused on multiple risk factor control (MRFC) in patients with both diabetes and CKD in a wide clinical practice setting. Generally, patients with kidney disease have been under-represented in cardiovascular clinical trials.[13] This population may have an altered risk-benefit profile, and extrapolation of data based on patients with normal kidney function into patients with CKD may be unreliable.[13] We aimed to conduct a pragmatic evaluation of the effectiveness of MRFC on mortality and cardiovascular events in participants with type 2 diabetes and CKD in a population-based cohort study.

## METHODS

### Data sources

This study employed a linked dataset derived from the UK Clinical Practice Research Datalink (CPRD), the UK National Health Service Hospital Episodes Statistics (HES) inpatient data and the UK Office for National Statistics (ONS) mortality data. The CPRD contains anonymised electronic health records from general practices across the UK.[14] The CPRD collects data for diagnoses and clinical assessment, prescriptions and laboratory test results, such as Haemoglobin A1c (HbA1c) and serum creatinine. The HES inpatient data were comprised inpatient records from all National Health Service hospitals in England. Information on the date of death and the causes of death was available in the ONS mortality data file. Multiple causes of death can be recorded in the mortality data. Diagnoses and clinical evaluation in the CPRD were coded with the Read codes, a hierarchical coding system used in primary care in the UK, whereas those in the HES and ONS were coded with the International Classification of Diseases, 10th Revision (ICD-10). Linked data are available for general practices in England only and participants were limited to those with linked data for the HES and ONS available.

### Study population

The scheme of the study cohort selection is presented in online supplementary figure S1. We initially sampled participants who were diagnosed with type 2 diabetes from the CPRD.[15] Using the CPRD records, the date of the first valid serum creatinine value between 2006 and 2010 recorded more than 1 year after the first diagnosis of diabetes were defined as the index date. A similar approach was taken by Adamsson Eryd et al[16] to ensure that participants managed for diabetes had sufficient time available for recording of baseline values. To avoid misclassification of CKD status and stage, the index serum creatinine values were validated by confirmation of subsequent values within 30% of the index values. We restricted

the sample to participants aged 40–79 years at the index date with at least 1 year of follow-up data available (ie, participants who died in the first year of follow-up were excluded). Estimated glomerular filtration rate (eGFR) was calculated from a serum creatinine value, age, gender and ethnicity, using the CKD Epidemiology Collaboration equation.[17] Missing ethnicity was assumed as 'non-black' in the present study. Participants diagnosed with end-stage renal disease, those who had received renal replacement therapy, or those with index eGFR $<15 \,\mathrm{mL/min/1.73 \,m^2}$ were excluded. We also excluded participants with missing data for smoking status, body mass index (BMI), HbA1c, blood pressure or total cholesterol, or those with extreme BMI ($<18.5$ or $\geq 45 \,\mathrm{kg/m^2}$) at baseline. Since it has been reported that low values of cardiovascular risk factors were not always associated with better outcomes in observational studies,[15 18 19] possibly due to reverse causation,[20 21] participants with low HbA1c ($<42 \,\mathrm{mmol/mol}$ or $<6.0\%$), blood pressure (systolic $<120$ or diastolic $<60 \,\mathrm{mm \,Hg}$) and total cholesterol ($<3 \,\mathrm{mmol/L}$) were further excluded. Participants were categorised according to index eGFR into participants with CKD ($<60 \,\mathrm{mL/min/1.73 \,m^2}$) and those with non-CKD ($\geq 60 \,\mathrm{mL/min/1.73 \,m^2}$).

### Multiple risk factor control

MRFC was defined in this study as consisting of four components: (1) HbA1c $<53 \,\mathrm{mmol/mol}$ ($<7.0\%$), (2) blood pressure $<140/90 \,\mathrm{mm \,Hg}$ (systolic $<140$ and diastolic $<90 \,\mathrm{mm \,Hg}$), (3) total cholesterol $<5 \,\mathrm{mmol/L}$ and (4) no smoking (non-smokers or ex-smokers). The means of HbA1c, blood pressure and total cholesterol records within 1 year before the index date were evaluated. The number of the risk factors controlled from four criteria was treated as the exposure and included as a categorical variable in the main analyses, with those meeting no or one criterion as a reference category.

### Outcomes

Main outcomes of interest in this study included all-cause and cardiovascular mortality, fatal and non-fatal coronary heart disease (CHD) and stroke. The date of death and causes of death were determined using the ONS mortality data. Participants who died from cardiovascular causes were identified if people had any of the ICD-10 codes I00–I99 as a cause of death. Similarly, participants who died from renal causes were identified by the ICD-10 codes N17–N19. All of the CPRD, HES and ONS were used to ascertain fatal and non-fatal CHD and stroke. Read codes for CHD and stroke reported previously[22 23] were updated for the present study. The ICD-10 codes for CHD and stroke were I20–I25 and I60–I64, respectively.

### Analysis

Baseline characteristics of the study cohort were described according to CKD status. Time-to-event analyses were conducted to evaluate the associations of MRFC with mortality and cardiovascular events. To address the issue of reverse causation and to avoid misclassification

of the outcomes from those which had existed at baseline, person-years for participants who experienced outcomes of interest in the first year of follow-up were excluded from analyses (online supplementary figure S1). Cox proportional hazards models were used to evaluate the association of MRFC with all-cause mortality. Proportional hazards assumption was assessed by visual inspection of log–log plots, and no apparent violation was found. Competing risks regression with subdistribution hazard models were conducted for cardiovascular mortality and cardiovascular events, considering competing risks for non-cardiovascular and all-cause death, respectively.[24] Associations of MRFC with cardiovascular events were evaluated in participants without a known history of cardiovascular diseases at baseline (online supplementary figure S1). Participants were followed from the index date until the earliest of the events of interest, the last date of CPRD records or 31 March 2015 for all-cause mortality evaluation. In the competing risks regression analyses for cardiovascular mortality and cardiovascular events, participants who experienced the corresponding competing events prior to the event of interest were censored.

Main analyses were conducted by CKD status, adjusting for a range of baseline covariates, including age (continuous), gender (male or female), CKD stage (3a, 3b and 4; for CKD cohort), BMI (18.5–24.9, 25.0–29.9, 30.0–34.9, 35.0–39.9 and 40.0–44.9 kg/m²), deprivation level (quintile; 1, least deprived, to 5, most deprived), duration of diabetes (1.0–4.9, 5.0–9.9 and 10+ years), proteinuria status, including microalbuminuria (yes, no and a missing category), a history of cardiovascular diseases, including CHD and stroke (for mortality evaluation), and prescribing during 6 months prior to the index date of antidiabetic drugs (none, insulin with and without other antidiabetic drugs, and non-insulin drugs only), antihypertensive drugs (none, drugs acting on renin–angiotensin system with and without other antihypertensive drugs, and other classes of antihypertensive drugs only, including β-blockers, calcium channel blockers and thiazide diuretics), statins and antiplatelet drugs, and index year (2006–2010). In addition, the associations of CKD with the outcomes were evaluated according to the number of risk factors controlled, adjusting for the potential confounding factors.

In this paper, the results for participants with CKD were focused on, with the results for those with non-CKD shown for comparative purposes. The associations of each component of MRFC with the outcomes were also evaluated to aid interpretation of the study results. All analyses were performed using Stata V.14 (Stata Corp). The 'forestplot' package in R was used to present the results.[25]

## Patient and public involvement

No patients were involved in setting the research question or the outcome measures, nor were they involved in developing plans for design or implementation of the study. No patients were asked to advise on interpretation or writing up of results. Results will be disseminated to relevant patient communities through news media.

# RESULTS

## Characteristics of the study population

Baseline characteristics of the study cohort are shown according to CKD status in table 1. Mean index eGFR was 49 mL/min/1.73 m² for participants with CKD and 81 mL/min/1.73 m² for those with non-CKD. Participants with CKD were older (71 vs 62 years), included more women (52% vs 40%), had a longer duration of diabetes and were more likely to have a history of cardiovascular diseases (37% vs 22%). A higher frequency of proteinuria was recorded in participants with CKD (18% vs 12% among participants with records of proteinuria status). HbA1c and total cholesterol were slightly lower in participants with CKD. Although diastolic blood pressure was lower in participants with CKD, systolic blood pressure was higher despite more people under antihypertensive medications. Participants with CKD were prescribed insulin, drugs on renin–angiotensin system, statins and antiplatelet drugs more frequently.

## Implementation of MRFC

A number of risk factors controlled from four components of MRFC are shown in table 2. More detailed results of which of the components were controlled are available in online supplementary table S1. Higher rates of control for HbA1c, total cholesterol and smoking status were observed in participants with CKD compared with those with non-CKD. However, blood pressure was less likely managed in participants with CKD (46% vs 51%). There were some differences in management status according to a history of cardiovascular diseases; in participants with CKD, higher rates of control of blood pressure (49% vs 44%) and total cholesterol (83% vs 76%) in participants with a history of cardiovascular diseases compared with those without. Participants meeting three or four criteria accounted for 37% or 13% in participants with CKD.

## Effectiveness of MRFC

Absolute risks for mortality and cardiovascular diseases and adjusted relative hazards of the number of risk factors controlled for the outcomes are shown in figure 1. Increasing numbers of risk factors controlled were associated with lower relative hazards for all outcomes studied relative to participants meeting no or one criterion. For participants with CKD meeting four MRFC criteria, the adjusted HR for all-cause mortality was 0.60 (95% CI 0.53 to 0.69) and adjusted subdistribution HR for cardiovascular mortality was 0.60 (95% CI 0.50 to 0.70). Participants meeting four criteria also had lower relative risks for CHD (adjusted subdistribution HR 0.73, 95% CI 0.59 to 0.91) and stroke (0.63, 95% CI 0.45 to 0.89) in participants with CKD. In participants with non-CKD, increasing numbers of risk factors controlled were also associated with lower risks for all-cause and cardiovascular mortality, CHD and

**Table 1**  Baseline characteristics of the study cohort by CKD status

| | CKD (n=11 431) | Non-CKD (n=36 429) | P values |
|---|---|---|---|
| Age (years) | | | |
| Mean (SD) | 71 (6) | 62 (9) | <0.001 |
| Gender | | | |
| Male | 5481 (48) | 22 006 (60) | <0.001 |
| Female | 5950 (52) | 14 423 (40) | |
| eGFR (mL/min/ 1.73 m$^2$) | | | |
| Mean (SD) | 49 (9) | 81 (13) | – |
| 15–29 | 558 (5) | – | |
| 30–44 | 2655 (23) | – | |
| 45–59 | 8218 (72) | – | |
| Smoking status | | | |
| Non-smoker | 5426 (47) | 16 511 (45) | <0.001 |
| Ex-smoker | 4327 (38) | 12 217 (34) | |
| Current smoker | 1678 (15) | 7701 (21) | |
| BMI (kg/m$^2$) | | | |
| 18.5–24.9 | 1459 (13) | 4097 (11) | <0.001 |
| 25.0–29.9 | 4329 (38) | 13 054 (36) | |
| 30.0–34.9 | 3527 (31) | 11 485 (32) | |
| 35.0–39.9 | 1541 (13) | 5454 (15) | |
| 40.0–44.9 | 575 (5) | 2339 (6) | |
| Deprivation level (quintile) | | | |
| 1 (least deprived) | 1508 (13) | 4785 (13) | 0.293 |
| 2 | 2331 (20) | 7300 (20) | |
| 3 | 2374 (21) | 7640 (21) | |
| 4 | 2637 (23) | 8172 (22) | |
| 5 (most deprived) | 2581 (23) | 8532 (23) | |
| Duration of diabetes (years) | | | |
| 1.0–4.9 | 5208 (46) | 22 527 (62) | <0.001 |
| 5.0–9.9 | 2954 (26) | 8356 (23) | |
| ≥10.0 | 3269 (29) | 5546 (15) | |
| Proteinuria | | | |
| Yes | 1714 (15) | 3279 (9) | <0.001 |
| No | 7666 (67) | 24 110 (66) | |
| Missing | 2051 (18) | 9040 (25) | |
| History of coronary heart disease and/or stroke | 4215 (37) | 7860 (22) | <0.001 |
| HbA1c (mmol/mol or %) | | | |
| 42–47 (6.0–6.4)* | 1307 (11) | 3513 (10) | <0.001 |
| 48–52 (6.5–6.9) | 3041 (27) | 8900 (24) | |
| 53–57 (7.0–7.4) | 2590 (23) | 7781 (21) | |
| 58–63 (7.5–7.9) | 1709 (15) | 5461 (15) | |
| 64–68 (8.0–8.4) | 1038 (9) | 3567 (10) | |
| ≥69 (≥8.5) | 1746 (15) | 7207 (20) | |

Continued

**Table 1**  Continued

| | CKD (n=11 431) | Non-CKD (n=36 429) | P values |
|---|---|---|---|
| Systolic blood pressure (mm Hg) | | | |
| 120–129 | 1777 (16) | 7203 (20) | <0.001 |
| 130–139 | 3508 (31) | 12 121 (33) | |
| 140–149 | 3387 (30) | 10 242 (28) | |
| ≥150 | 2759 (24) | 6863 (19) | |
| Diastolic blood pressure (mm Hg) | | | |
| 60–79 | 7238 (63) | 16 803 (46) | <0.001 |
| 80–89 | 3599 (31) | 15 816 (43) | |
| ≥90 | 594 (5) | 3810 (10) | |
| Total cholesterol (mmol/L) | | | |
| 3.0–3.9 | 3782 (33) | 10 960 (30) | <0.001 |
| 4.0–4.9 | 5220 (46) | 16 387 (45) | |
| ≥5.0 | 2429 (21) | 9082 (25) | |
| Medication | | | |
| Antidiabetic drugs | | | <0.001 |
| Insulin (±non-insulin) | 1805 (16) | 3225 (9) | |
| Non-insulin only | 7722 (68) | 26 753 (73) | |
| Antihypertensive drugs | | | <0.001 |
| Drugs on renin–angiotensin system (±others) | 8472 (74) | 21 535 (59) | |
| Other antihypertensive drugs only | 1610 (14) | 4751 (13) | |
| Statins | 9004 (79) | 27 011 (74) | <0.001 |
| Antiplatelet drugs | 6440 (56) | 16 375 (45) | <0.001 |
| Index years | | | |
| 2006 | 9091 (80) | 24 192 (66) | <0.001 |
| 2007 | 1008 (9) | 3741 (10) | |
| 2008 | 545 (5) | 2880 (8) | |
| 2009 | 432 (4) | 2677 (7) | |
| 2010 | 355 (3) | 2939 (8) | |

Frequencies (percentages) are shown otherwise specified.
*Participants with HbA1c <48 mmol/mol (<6.5%) were only included if they were prescribed antidiabetic drugs.
BMI, body mass index; CKD, chronic kidney disease; eGFR, estimated glomerular filtration rate.

stroke. As shown in online supplementary figure S2, the strengths of associations of each component of MRFC with mortality and cardiovascular diseases were different; for example, the greatest associations of no smoking with all-cause and cardiovascular mortality were observed in participants with and without CKD.

### Comparisons between CKD and non-CKD

Unadjusted absolute risks for mortality and cardiovascular diseases were higher in participants with CKD by 1.4-fold to 2.9-fold compared with those with non-CKD at the same MRFC category (figure 1). More participants

**Table 2** Risk factors controlled according to CKD and a history of CVD

| | CKD | | | Non-CKD | | |
|---|---|---|---|---|---|---|
| | Total (n=11 431) | No CVD (n=7216) | CVD (n=4215) | Total (n=36 429) | No CVD (n=28 569) | CVD (n=7860) |
| **Individual risk factor controlled** | | | | | | |
| HbA1c <53 mmol/mol (<7.0%) | 4348 (38) | 2767 (38) | 1581 (38) | 12 413 (34) | 9603 (34) | 2810 (36) |
| Blood pressure <140 and <90 mm Hg | 5224 (46) | 3147 (44) | 2077 (49) | 18 655 (51) | 14 438 (51) | 4217 (54) |
| Total cholesterol <5 mmol/L | 9002 (79) | 5512 (76) | 3490 (83) | 27 347 (75) | 20 826 (73) | 6521 (83) |
| No smoking | 9753 (85) | 6193 (86) | 3560 (84) | 28 728 (79) | 22 565 (79) | 6163 (78) |
| **No of risk factors controlled** | | | | | | |
| 0 | 138 (1) | 87 (1) | 51 (1) | 806 (2) | 678 (2) | 128 (2) |
| 1 | 1427 (12) | 971 (13) | 456 (11) | 5372 (15) | 4421 (15) | 951 (12) |
| 2 | 4162 (36) | 2693 (37) | 1469 (35) | 13 288 (36) | 10 602 (37) | 2686 (34) |
| 3 | 4240 (37) | 2598 (36) | 1642 (39) | 12 657 (35) | 9665 (34) | 2992 (38) |
| 4 | 1464 (13) | 867 (12) | 597 (14) | 4306 (12) | 3203 (11) | 1103 (14) |

Frequencies (percentages) are shown.
CKD, chronic kidney disease; CVD, (a history of) cardiovascular diseases.

with CKD died from cardiovascular causes compared with those without (63% vs 54%, p<0.001). More participants with CKD died from renal causes (n=631 or 5% vs n=326 or 0.9%, p<0.001), but the proportions were much smaller than cardiovascular causes of death. Relative hazards of CKD for the outcomes are shown in figure 2. After adjustment with possible confounding factors, comorbid CKD remained to be associated with greater risks for all-cause mortality (adjusted HR 1.16 to 1.30) and cardiovascular mortality (adjusted subdistribution HR 1.25 to 1.41). In participants meeting two or more criteria, comorbid CKD was associated with a higher risk for CHD (1.18 to 1.25). The associations of comorbid CKD with stroke were observed in participants meeting four criteria only (1.64).

## DISCUSSION

In this population-based cohort study of participants with type 2 diabetes and CKD stages 3 or 4, MRFC was associated with lower relative risks for mortality and cardiovascular diseases. We also confirmed that CKD was associated with increased risks for mortality and cardiovascular events. Higher absolute risks for mortality and cardiovascular events and great relative risk reduction associated with MRFC suggest that the MRFC strategy may be one of the main approaches to potentially reducing the burden of diabetes and CKD.

This study evaluated the effectiveness of MRFC in patients with type 2 diabetes according to presence or absence of CKD in clinical practice. So far, the associations of MRFC with lower risks for mortality and cardiovascular events have been shown in people with diabetes, not focusing on CKD status. Participants with controlled three risk factors of HbA1c, blood pressure and low-density lipoprotein (LDL) cholesterol had 62% and 60% risk reduction for

cardiovascular events and CHD, respectively, in patients with diabetes without known cardiovascular diseases.[10] The associations of uncontrolled HbA1c, blood pressure, LDL cholesterol and smoking with mortality and cardiovascular events were individually evaluated in a large population-based study with >850 000 participants with diabetes.[11] The study cohort included 35.5% of participants with CKD in those with cardiovascular diseases and 21.8% in those without, and CKD was included in the analyses for adjustment. This study suggested that uncontrolled risk factors attributed to about 1 in 3 major cardiovascular events and fewer 1 in 10 deaths.

The strength of this study was the inclusion of a large size of >11 000 participants with diabetes and CKD with an observation of >62 000 person-years. In addition to the large sample size and long-term follow-up, representativeness from general population and data quality are also advantages of the CPRD,[14] which should remain even if linked data for the HES and ONS are only available for England practices. Instead, linked data for diagnoses in hospitals and death registration substantially enhanced the validity of the study to evaluate mortality and cardiovascular events.

There are also some limitations in this study. First, despite our focus on the number of MRFC, the impacts of each component of MRFC on mortality and cardiovascular events were different. Different cut-off points for HbA1c, blood pressure and total cholesterol may bring different results. Next, we could not conclude that associations represented causal relationships between MRFC and mortality and cardiovascular events in this non-randomised study. There is a possibility of confounding if healthier participants were managed more successfully and this resulted in being categorised as those with greater number of risk factors controlled.

**A  CKD**

| Number of risk factors controlled | N | Number of events | Observation (1,000 py) | Rate (per 1,000 py) | | Adjusted HR/SHR (95% CI) | P value |
|---|---|---|---|---|---|---|---|
| **[ All participants ]** | | | | | | | |
| **All-cause mortality** | | | | | | | |
| 4 | 1,464 | 388 | 8.1 | 48.1 | | 0.60 (0.53 to 0.69) | <0.001 |
| 3 | 4,240 | 1,301 | 23.2 | 56.0 | | 0.70 (0.63 to 0.77) | <0.001 |
| 2 | 4,162 | 1,393 | 22.6 | 61.5 | | 0.77 (0.70 to 0.85) | <0.001 |
| 0+1 | 1,565 | 598 | 8.2 | 72.5 | | Reference | |
| **Cardiovascular mortality** | | | | | | | |
| 4 | 1,464 | 236 | 8.1 | 29.3 | | 0.60 (0.50 to 0.70) | <0.001 |
| 3 | 4,240 | 821 | 23.2 | 35.3 | | 0.71 (0.62 to 0.80) | <0.001 |
| 2 | 4,162 | 893 | 22.6 | 39.4 | | 0.79 (0.70 to 0.90) | <0.001 |
| 0+1 | 1,565 | 372 | 8.2 | 45.1 | | Reference | |
| **[ Participants without previous CHD/stroke ]** | | | | | | | |
| **Coronary heart disease** | | | | | | | |
| 4 | 851 | 143 | 4.5 | 32.0 | | 0.73 (0.59 to 0.91) | 0.004 |
| 3 | 2,537 | 510 | 13.3 | 38.4 | | 0.86 (0.73 to 1.00) | 0.053 |
| 2 | 2,635 | 558 | 13.4 | 41.6 | | 0.88 (0.75 to 1.03) | 0.114 |
| 0+1 | 1,025 | 232 | 5.1 | 45.5 | | Reference | |
| **Stroke** | | | | | | | |
| 4 | 865 | 51 | 4.7 | 10.9 | | 0.63 (0.45 to 0.89) | 0.009 |
| 3 | 2,580 | 173 | 14.0 | 12.3 | | 0.72 (0.56 to 0.92) | 0.010 |
| 2 | 2,670 | 186 | 14.3 | 13.0 | | 0.72 (0.56 to 0.93) | 0.010 |
| 0+1 | 1,044 | 102 | 5.4 | 18.9 | | Reference | |

0.25  0.5  0.75  1  1.25
Adjusted HR/SHR

**B  Non-CKD**

| Number of risk factors controlled | N | Number of events | Observation (1,000 py) | Rate (per 1,000 py) | | Adjusted HR/SHR (95% CI) | P value |
|---|---|---|---|---|---|---|---|
| **[ All participants ]** | | | | | | | |
| **All-cause mortality** | | | | | | | |
| 4 | 4,306 | 515 | 23.3 | 22.1 | | 0.62 (0.55 to 0.69) | <0.001 |
| 3 | 12,657 | 1,670 | 69.7 | 23.9 | | 0.69 (0.63 to 0.74) | <0.001 |
| 2 | 13,288 | 1,928 | 73.4 | 26.3 | | 0.77 (0.71 to 0.83) | <0.001 |
| 0+1 | 6,178 | 935 | 33.3 | 28.1 | | Reference | |
| **Cardiovascular mortality** | | | | | | | |
| 4 | 4,306 | 266 | 23.3 | 11.4 | | 0.55 (0.47 to 0.64) | <0.001 |
| 3 | 12,657 | 912 | 69.7 | 13.1 | | 0.63 (0.57 to 0.70) | <0.001 |
| 2 | 13,288 | 1,015 | 73.4 | 13.8 | | 0.68 (0.61 to 0.75) | <0.001 |
| 0+1 | 6,178 | 545 | 33.3 | 16.4 | | Reference | |
| **[ Participants without previous CHD/stroke ]** | | | | | | | |
| **Coronary heart disease** | | | | | | | |
| 4 | 3,164 | 305 | 16.6 | 18.4 | | 0.54 (0.47 to 0.62) | <0.001 |
| 3 | 9,528 | 1,130 | 50.4 | 22.4 | | 0.65 (0.59 to 0.71) | <0.001 |
| 2 | 10,446 | 1,343 | 54.8 | 24.5 | | 0.70 (0.64 to 0.77) | <0.001 |
| 0+1 | 5,010 | 808 | 25.4 | 31.8 | | Reference | |
| **Stroke** | | | | | | | |
| 4 | 3,190 | 63 | 17.2 | 3.7 | | 0.37 (0.28 to 0.50) | <0.001 |
| 3 | 9,628 | 325 | 52.5 | 6.2 | | 0.63 (0.53 to 0.75) | <0.001 |
| 2 | 10,565 | 424 | 57.2 | 7.4 | | 0.75 (0.63 to 0.88) | <0.001 |
| 0+1 | 5,079 | 234 | 26.8 | 8.7 | | Reference | |

0.25  0.5  0.75  1  1.25
Adjusted HR/SHR

**Figure 1** Relative hazards of the number of risk factors controlled for mortality and cardiovascular events in (A) participants with chronic kidney disease (CKD) and (B) participants with non-CKD. HRs for all-cause mortality and subdistribution HRs (SHRs) for cardiovascular mortality, coronary heart disease (CHD) and stroke were adjusted for age, gender, CKD stage (for CKD cohort), body mass index, deprivation level, duration of diabetes, proteinuria status, a history of cardiovascular diseases (for mortality evaluation), prescribing of antidiabetic, antihypertensive, statins and antiplatelet drugs, and index year.

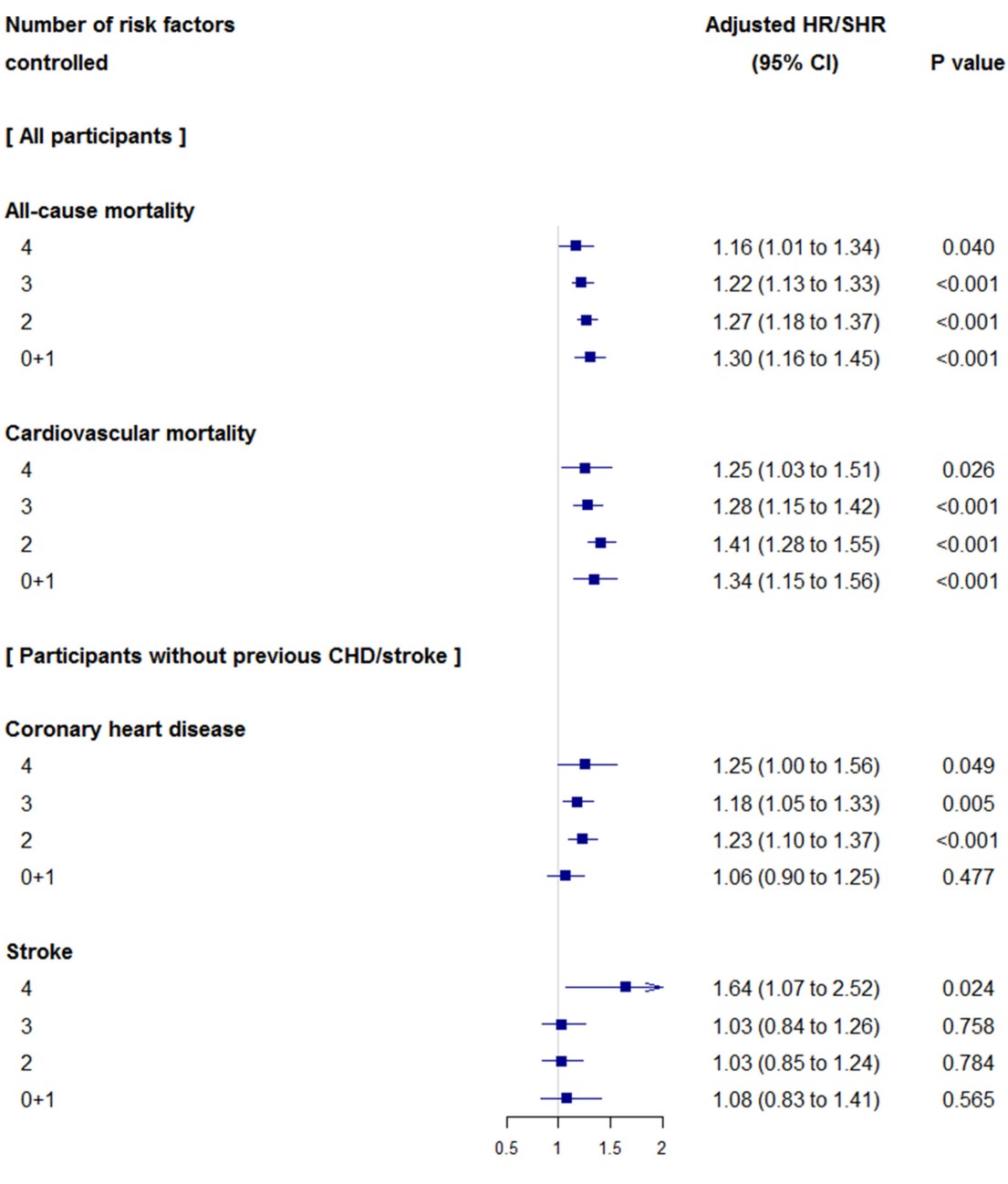

**Figure 2** Relative hazards of presence of chronic kidney disease (CKD) for mortality and cardiovascular events compared with non-CKD as reference. HRs for all-cause mortality and subdistribution HRs (SHRs) for cardiovascular mortality, coronary heart disease (CHD) and stroke were adjusted for age, gender, body mass index, deprivation level, duration of diabetes, proteinuria status, a history of cardiovascular diseases (for mortality evaluation), prescribing of antidiabetic, antihypertensive, statins and antiplatelet drugs, and index year.

For example, stringent management of HbA1c might not be targeted for vulnerable participants due to concerns for greater risk of hypoglycaemia, a form of confounding by contraindication. We cannot exclude the possibility of residual confounding, despite adjustment with a range of covariates in the analyses, including physical activity and alcohol intake.[26 27] Then, measurement and assay methods for HbA1c, blood pressure, cholesterol and serum creatinine might not have been standardised among general practices or laboratories.

As well as missing data on ethnicity and fluctuations in serum creatinine values, these methodological limitations might influence the determination of CKD status or staging. Although proteinuria has been known as a risk factor for mortality and cardiovascular diseases,[28 29] we could not determine proteinuria status completely as reported previously.[30 31] Incomplete records on proteinuria may introduce a bias for proteinuria status and possibly influence the study results. Finally, although we used one of the largest primary care electronic health

records database, it seemed to be insufficient to separately evaluate MRFC for participants with different stages of CKD. Further research is needed to focus on patients with more advanced CKD who may have altered risk-benefit profile compared with patients with less impaired renal function.

In summary, based on the population-based cohort study of routine clinical practices, MRFC may lower the increased risks for mortality and cardiovascular events in people with diabetes and CKD. Further research is needed to evaluate appropriateness of MRFC according to individual participants' health status for improved management of cardiovascular risks in this population.

**Contributors** Both authors contributed to conception and study design of the study, data acquisition, statistical analysis and interpretation. SH drafted the manuscript and MCG revised it critically for important intellectual content.

**Funding** SH was supported by the Uehara Memorial Foundation Postdoctoral Fellowship, Tokyo, Japan. MCG was supported by the National Institute for Health Research (NIHR) Biomedical Research Centre at Guy's and St Thomas' NHS Foundation Trust and King's College London. This study is based on data from the CPRD obtained under license from the UK Medicines and Healthcare products Regulatory Agency. However, the interpretation and conclusions contained in this report are those of the authors alone and not necessarily those of the National Health Service, the NIHR or the Department of Health. Open access for this article was funded by King's College London.

**Competing interests** None declared.

**Patient consent** Not required.

**Ethics approval** CPRD Independent Scientific Advisory Committee.

**Provenance and peer review** The study was approved by the CPRD Independent Scientific Advisory Committee (ISAC Protocol 15_201R).

**Data sharing statement** No additional data are available.

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
