## [Reviewer comments · BMJ Open]

ARTICLE DETAILS

TITLE (PROVISIONAL)	Multiple risk factor control, mortality and cardiovascular events in type 2 diabetes and chronic kidney disease: a population-based cohort study
AUTHORS	Hamada, Shota; Gulliford, Martin

VERSION 1 – REVIEW

REVIEWER	Hiroki Yokoyama Jiyugaoka Medical CL., Int. Med. Japan
REVIEW RETURNED	27-Oct-2017

GENERAL COMMENTS	Dr Hamada et al performed a study to explore the effect of MRFC on reducing CVD in diabetic patients with and without CKD. The study included a large-scale of participants (N=47,860) and a long follow-up period (6 years). This study is interesting in that incidence of CVD and mortality were presented in subjects with and without presence of CKD, separately. However, there are several concerns in the presentation and interpretation. 1) Authors performed this study to explore treatment approaches that leads to prevention of CVD in subjects with diabetes and CKD, as described in Introduction. They indicated that MRFC is associated with lower risk for developing CVD. But the comment in Discussion says that “We could not determine the causal relationships between MRFC and mortality and cardiovascular events from this non-randomised study.” I cannot understand what you mean. The data obtained from primary care database may reflect the real-world practice; am I right? 2) It is rather confusing that the main results of Figure 1 and 2 dealt subjects with no CVD to calculate incidence of coronary heart disease and stroke, and dealt all subjects to calculate mortality. I think authors should separate the presentation on subjects with and without a history of CVD. Based on this, authors can indicate the effect of MRFC in those with and without CKD. Similarly, the first sentence in Discussion is not correct. The sentence have to be; In this population-based cohort study of 11,431 participants FOR mortality study and 7,216 FOR CVD morbidity with type 2 diabetes and CKD stages 3 to 4, MRFC was associated with lower relative risks for mortality and cardiovascular diseases. 3) From a clinical point of view, it remains a question whether the number of MRFC can just discriminate the impact of risk on predicting CVD, and whether the impact of each four risk in MRFC is the same between subjects with and without a history of CVD. The association of MRFC with lower risk for developing CVD should be
--

	verified by not only the number of simple category but also considering different cut-off value in HbA1c, BP, and cholesterol. The impact of each risk factor on developing CVD has to be evaluated. 4) Authors described that “we found that the implementation of MRF in patients with diabetes was suboptimal”. This is vague and unclear. What is optimal and what is poor in your definition? 5) It is very well-known that the degree of albuminuria has a profound and strong impact on cardiovascular disease morbidity and mortality. This means that a person with controlled A1c and BP but has macroalbuminuria is quite at high risk for cardiovascular disease morbidity. I think the present result would be affected by incorporating the degree of albuminuria in the analysis. Please keep in mind that ALBUMINURIA is a MAJOR component of CKD. Table 1 and 2 “Parenthesis indicates percentage of the subjects.” should be given.
--	---

REVIEWER	Mariko Miyazaki Tohoku University, Graduate School of Medicine JAPAN
REVIEW RETURNED	13-Nov-2017

GENERAL COMMENTS	I’m glad to have the opportunity to review this article. This study revealed the significance of multiple risk factor control by primary care against cardiovascular mortality in UK. I live in out of UK, however, deeply interested in the investigation. I have some comments as bellows:.  1. The reason why lower value of some factors excluded in this analysis should be described more detail in discussion section. It is explained in the methods with several reference, however, the comparison with SPRINT study may be important to evaluate the risk in the lower range of blood pressure. 2. The risk of overweight is little mentioned in this study. How was the Hazard Ratio in single regression compared with 1st quantile. 3. Have the number of risk of renal death as non-cardiovascular mortality impacted in CKD subgroup ? Overall, I would suggest that this article needs minor revision to be published in BMJ Open..
--

REVIEWER	Russell de Souza McMaster University, Hamilton, ON, Canada
REVIEW RETURNED	14-Dec-2017

GENERAL COMMENTS	In this prospective study, the authors assess the association of optimal multiple risk factor control (cholesterol, glycemia, tobacco smoke, and blood pressure) on all-cause and cardiovascular mortality in patients with type 2 diabetes with and without chronic kidney disease. Overall, the analyses are well performed, and the conclusions justified-- but I think that the modeling approach needs a revisit. As described, it was not optimal to address what appears to be the most interesting finding of the study. Specific comment:  1. In the "Strengths and limitations" piece (pages 3-4), I was confused by the 4th bullet. My understanding of confounding by
--

	indication is that a variable is a risk factor for a disease among nonexposed persons and is associated with the exposure of interest in the population from which the cases derive. It is also not an intermediate step in the causal pathway between the exposure and the disease. Is control of the risk factors the mechanism by which CV risk is modulated by treatment? If so, then I think CBI is not quite what is happening here. I apologize if I have misunderstood this, feel free to correct me if so. 2. I felt that there was a more appropriate way to analyze the data. The paper reads as if the goal is to see if having CKD modified the cardiovascular risk of T2DM. if so, should the correct approach not have been to run the Cox-PH model with 1) optimal control; 2) CKD; 3) optimal control x CKD as your set of X (independent) predictors; and have your DV be your cardiovascular outcomes? Then you could produce Kaplan-Meier curves comparing the rates of CVD in those who are optimally controlled , and see if the risk is modified by CKD? $Y_{ij} = A + B1(\text{optimal control}) + B2(\text{cKD}) + B3 (\text{optimal x CKD}) + e_{ij}$ This I think is a stronger test of your primary hypothesis.
--	---

VERSION 1 – AUTHOR RESPONSE

Responses to the Reviewers' comments:

For Reviewer: 1

1) Authors performed this study to explore treatment approaches that leads to prevention of CVD in subjects with diabetes and CKD, as described in Introduction. They indicated that MRFC is associated with lower risk for developing CVD. But the comment in Discussion says that "We could not determine the causal relationships between MRFC and mortality and cardiovascular events from this non-randomised study." I cannot understand what you mean. The data obtained from primary care database may reflect the real-world practice; am I right?

Thank you for your comment. We now say 'we could not conclude that associations represented causal relationships between MRFC and mortality and cardiovascular events in this non-randomised study'. (Strengths and limitations of this study, page 4; Discussion, page 17)

This study included a broad range of participants with type 2 diabetes in the real-world clinical setting in the UK. This study has an advantage of 'generalisability' or 'external validity' given that most of patients with diabetes are managed in primary care. However, as we described, we could not establish the causal relationship (ie, 'internal validity') from this non-randomised observational study although we attempted to adjust various confounding factors in the analyses. Generally, randomized controlled trials or observational studies with quasi-experimental designs, when appropriate, are required to draw causal relationships. In this case, MRFC was not randomly assigned to participants. MRFC might be the result from adjustment of treatment intensity according to participants' health condition and other factors. Also, we could not the exclude the possibility of residual confounding. These can be barriers to establish the causal relationships between MRFC and lower risks for mortality and cardiovascular diseases.

2) It is rather confusing that the main results of Figure 1 and 2 dealt subjects with no CVD to calculate incidence of coronary heart disease and stroke, and dealt all subjects to calculate mortality. I think

authors should separate the presentation on subjects with and without a history of CVD. Based on this, authors can indicate the effect of MRFC in those with and without CKD.

Thank you very much for your suggestion. We now specify the cohorts to be subject to evaluate mortality and cardiovascular diseases separately in the figures (figures 1, 2, and S2). (Discussion, page 15)

Similarly, the first sentence in Discussion is not correct. The sentence have to be; In this population-based cohort study of 11,431 participants FOR mortality study and 7,216 FOR CVD morbidity with type 2 diabetes and CKD stages 3 to 4, MRFC was associated with lower relative risks for mortality and cardiovascular diseases.

Thank you. We have also revised the first sentence in the Discussion section based on your comment.

Discussion (page 14)

“In this population-based cohort study of participants with type 2 diabetes and CKD stages 3 to 4, MRFC was associated with lower relative risks for mortality (N>11,000) and cardiovascular diseases (N>7,000).”

3) From a clinical point of view, it remains a question whether the number of MRFC can just discriminate the impact of risk on predicting CVD, and whether the impact of each four risk in MRFC is the same between subjects with and without a history of CVD. The association of MRFC with lower risk for developing CVD should be verified by not only the number of simple category but also considering different cut-off value in HbA1c, BP, and cholesterol. The impact of each risk factor on developing CVD has to be evaluated.

Thank you very much for your valuable comments. Although evaluation of each component of MRFC was outside the scope of this study, we estimated the associations of individual risk factors on the outcomes and presented in supplementary data (figure S2). Accordingly, we have revised the manuscript as below.

Methods – Analysis (page 12)

“The associations of each component of MRFC with the outcomes were also evaluated to aid interpretation of the study results.”

Results – Effectiveness of MRFC (page 14)

“As shown in figure S2, the strengths of associations of each component of MRFC with mortality and cardiovascular diseases were different; for example, the greatest associations of no smoking with all-cause and cardiovascular mortality were observed in participants with and without CKD.”

Discussion (page 17)

“There are also some limitations in this study. First, despite our focus on the number of MRFC, the impacts of each component of MRFC on mortality and cardiovascular events were different. Different cut-off points for HbA1c, blood pressure, and total cholesterol may bring different results.”

4) Authors described that “we found that the implementation of MRFC in patients with diabetes was suboptimal”. This is vague and unclear. What is optimal and what is poor in your definition?

Thank you very much for your comment for interpretation of our results. We could not say the implementation of MRFC was suboptimal based on this non-randomised study. We have removed the relevant descriptions in the revised manuscript.

We now say (Abstract, page 3; the second sentence in Discussion, page 18): “MRFC may lower increased risks for mortality and cardiovascular events in people with diabetes and CKD. Further research is needed to evaluate appropriateness of MRFC according to individual participants’ health status for improved management of cardiovascular risks in this population.”

5) It is very well-known that the degree of albuminuria has a profound and strong impact on cardiovascular disease morbidity and mortality. This means that a person with controlled A1c and BP but has macroalbuminuria is quite at high risk for cardiovascular disease morbidity. I think the present result would be affected by incorporating the degree of albuminuria in the analysis. Please keep in mind that ALBUMINURIA is a MAJOR component of CKD.

Thank you very much for your valuable suggestion. We now included the distribution of proteinuria status, including microalbuminuria, at baseline in table 1. We also conducted all of regression analyses adjusted for proteinuria status as one of major covariates to evaluate mortality and cardiovascular diseases morbidity. Hazard ratios obtained from the current analyses were slightly affected, but the conclusions of this study have not changed.

Methods – Analysis (page 11):

“Main analyses were conducted by CKD status, adjusting for a range of baseline covariates, including ..., duration of diabetes (1.0–4.9 5.0–9.9 and 10+ years), proteinuria status, including microalbuminuria (yes, no, and a missing category), ...”

Results – Characteristics of the study population (page 12):

“A higher frequency of proteinuria was recorded in participants with CKD (18% vs 12% among participants with proteinuria status)”

Table 1 and 2 “Parenthesis indicates percentage of the subjects.” should be given.

Thank you. We added the footnote based on your suggestion.

For Reviewer: 2

1. The reason why lower value of some factors excluded in this analysis should be described more detail in discussion section. It is explained in the methods with several reference, however, the comparison with SPRINT study may be important to evaluate the risk in the lower range of blood pressure.

Thank you very much for your suggestions on this important point in methodology. Taking into account previous findings on potential reverse causation observed in some epidemiological studies, we favour keeping the current approach to selecting the study cohort. However, this may introduce selection bias and we now added some discussion related to this as below.

Discussion (page 17)

“We excluded participants with low values for HbA1c, blood pressure, or total cholesterol in order to minimise the potential for reverse causation observed in epidemiological studies as described before, which might introduce selection bias of the study population.”

2. The risk of overweight is little mentioned in this study. How was the Hazard Ratio in single regression compared with 1st quantile.

Thank you for your suggestion. We acknowledge body weight is one of the major cardiovascular risk factors, and included this as BMI (18.5–24.9, 25.0–29.9, 30.0–34.9, 35.0–39.9 and 40.0–44.9 kg/m²) in the analyses. Please note that we excluded participants with extreme BMI (ie, <18.5 or ≥45 kg/m²) in this study as described in the Methods section. In this study, we focused on four risk factors, namely HbA1c, blood pressure, total cholesterol and smoking status. Although body weight was associated with mortality and cardiovascular diseases, we would like to keep BMI as one of covariates in analyses and do not show hazard ratios for BMI specifically.

3. Have the number of risk of renal death as non-cardiovascular mortality impacted in CKD subgroup ?

Thank you for your comment. Distribution of renal causes of deaths between participants with CKD and non-CKD would be informative to understand the importance of cardiovascular risks reduction based on this study. We examined the causes of deaths using the ONS mortality data again. We identified 631 (5%) and 326 (0.9%) participants with CKD and non-CKD, respectively, had renal causes of deaths (ICD-10: N17 to N19) listed as a cause of deaths. The data is consistent with previous findings that people with diabetes were more likely to die before renal causes of deaths.

Methods – Outcomes (page 9):

“Similarly, participants who died from renal causes were identified by the ICD-10 codes N17 to N19.”

Results – Comparisons between CKD and non-CKD (page 14).

“More participants with CKD died from renal causes (n=631 or 5% vs n=326 or 0.9%, P<0.001), but the proportions were much smaller than cardiovascular causes of death.”

For Reviewer: 3

1. In the "Strengths and limitations" piece (pages 3-4), I was confused by the 4th bullet. My understanding of confounding by indication is that a variable is a risk factor for a disease among nonexposed persons and is associated with the exposure of interest in the population from which the cases derive. It is also not an intermediate step in the causal pathway between the exposure and the disease. Is control of the risk factors the mechanism by which CV risk is modulated by treatment? If so, then I think CBI is not quite what is happening here. I apologize if I have misunderstood this, feel free to correct me if so.

Thank you very much, we now explain as below.

Strengths and limitations of this study (page 4):

“There is a possibility of confounding if healthier participants were managed more successfully and this resulted in being categorised as those with greater number of risk factors controlled.”

Discussion (page 17):

“There is a possibility of confounding if healthier participants were managed more successfully and this resulted in being categorised as those with greater number of risk factors controlled. For example, stringent management of HbA1c might not be targeted for vulnerable participants due to concerns for greater risk of hypoglycaemia, a form of confounding by contra-indication.”

2. I felt that there was a more appropriate way to analyze the data. The paper reads as if the goal is to see if having CKD modified the cardiovascular risk of T2DM. If so, should the correct approach not have been to run the Cox-PH model with 1) optimal control; 2) CKD; 3) optimal control x CKD as your set of X (independent) predictors; and have your DV be your cardiovascular outcomes? Then you

could produce Kaplan-Meier curves comparing the rates of CVD in those who are optimally controlled , and see if the risk is modified by CKD?

$$Y_{ij} = A + B1(\text{optimal control}) + B2(\text{cKD}) + B3 (\text{optimal} \times \text{CKD}) + e_{ij}$$

This I think is a stronger test of your primary hypothesis.

Thank you very much for your suggestions. The approach you suggest is more appropriate to show the differences in risk reduction by MRFC in participants with CKD and non-CKD. However, we did not focus on the effect modification by CKD status in this study. We would like to show the risk reduction by MRFC in the sub-population with diabetes and CKD. Therefore, we would like to keep the current approach which investigated the associations of MRFC with the outcomes separately according to CKD status.

VERSION 2 – REVIEW

REVIEWER	Hiroki Yokoyama Jiyugaoka Medical Clinic, Japan
REVIEW RETURNED	19-Jan-2018

GENERAL COMMENTS	It is questionable that proportion of proteinuria missing is high even they collected data from inpatient records (usually inpatients should undergo many examinations, that would surely include albuminuria and retinopathy for patients with type 2 diabetes) and proportion of proteinuria (\geqmicroalbuminuria) is extremely low in subjects not only with CKD but also with non-CKD. Given that the proportion of missing was added to proteinuria YES, the proportion of proteinuria is still very low if compared to lots of the previous studies reporting the prevalence of KDIGO-classified groups by eGFR and albuminuria in type 2 diabetes. Albuminuria is a major determinant of cardiovascular morbidity and mortality. Now albuminuria is even a treatment marker to predict the prognosis in type 2 diabetes (J Am Soc Nephrol 2015;26:2055-64, RENAAL study, etc.). And a lot of studies have indicated that albuminuria and eGFR are significantly and independently associated with onset of cardiovascular disease and death. Concept of CKD is based on GFR and albuminuria, and this paper lacks a reliable data of albuminuria. This is a big drawback. I think authors have to reexamine albuminuria very carefully. The reason for missing should be described. This paper may be rejected for publication, or asked for another reviewer being familiar with diabetic kidney disease, or rewritten after careful reexamination of albuminuria, or rewritten after verification of methodology lacking for albuminuria.
---

REVIEWER	Russell J de Souza McMaster University, Canada
REVIEW RETURNED	12-Feb-2018

GENERAL COMMENTS	Thank you. You have provided sufficient rebuttal to my comments.
--

REVIEWER	Mariko Miyazaki Department of Nephrology, Endocrinology and Vascular medicine Tohoku University Graduate School of Medicine JAPAN
REVIEW RETURNED	12-Feb-2018

GENERAL COMMENTS	The authors addressed adequately to my comments and other reviewers. The manuscript has been revised well. I think this manuscript will be acceptable.
--

VERSION 2 – AUTHOR RESPONSE

Responses to the Reviewers' comments:

For Reviewer: 1

1) It is questionable that proportion of proteinuria missing is high even they collected data from inpatient records (usually inpatients should undergo many examinations, that would surely include albuminuria and retinopathy for patients with type 2 diabetes) and proportion of proteinuria (\geq microalbuminuria) is extremely low in subjects not only with CKD but also with non-CKD. Given that the proportion of missing was added to proteinuria YES, the proportion of proteinuria is still very low if compared to lots of the previous studies reporting the prevalence of KDIGO-classified groups by eGFR and albuminuria in type 2 diabetes.

Albuminuria is a major determinant of cardiovascular morbidity and mortality. Now albuminuria is even a treatment marker to predict the prognosis in type 2 diabetes (J Am Soc Nephrol 2015;26:2055-64, RENAAL study, etc.). And a lot of studies have indicated that albuminuria and eGFR are significantly and independently associated with onset of cardiovascular disease and death.

Concept of CKD is based on GFR and albuminuria, and this paper lacks a reliable data of albuminuria. This is a big drawback. I think authors have to reexamine albuminuria very carefully. The reason for missing should be described. This paper may be rejected for publication, or asked for another reviewer being familiar with diabetic kidney disease, or rewritten after careful reexamination of albuminuria, or rewritten after verification of methodology lacking for albuminuria.

Thank you very much for your valuable comments. We have confirmed the identification process of proteinuria testing and the test results again. We also cite some references to address a concern on the low recording rate and prevalence of proteinuria in the present study. However, as the reviewer indicated, and as discussed in the references, the prevalence of proteinuria may be underestimated in the present study, which has been added as one of the limitations of this study.

Recording of proteinuria status:

As the reviewer indicated, evaluation of proteinuria status is one of the key process measures in management of diabetes¹⁾. Nevertheless, in the UK, the recording rates of proteinuria status are about 75% with a drop in recent years to two-thirds according to some literature and the National Diabetes Audit¹⁻³⁾. The recording rate of proteinuria status in the present study was 77% in overall participants which is similar with previous reports. However, we cannot exclude the possibility of missing not at random for proteinuria testing, which may introduce a bias for proteinuria status.

We should have stated the HES data, an inpatient data, more clearly. The HES dataset used for this study included diagnostic and discharge data, but no test results. Thus, no additional information on proteinuria status were available from the HES data. Please also note that diabetes is generally managed in primary care in the UK, and therefore most of test results should be included in the CPRD, a primary care electronic health records database.

Prevalence of proteinuria:

In Table 1, we show the prevalence of proteinuria was 15% for participants with diabetes and CKD and 9% for those with non-CKD. When excluding participants with missing data on proteinuria status, these figures increased to 18% and 12%, respectively, as described in the main text (page 12).

We found some studies which reported proteinuria status in people with diabetes using primary care data in the UK. One study showed that the prevalence of proteinuria were 8.6% for participants with diabetes and 18.6% for participants with diabetes and CKD (eGFR < 60 mL/min/1.73 m²)². A more recent study using the same database with our study reported that the one-year prevalence of proteinuria was 12.3% after initiation of antidiabetic drugs³. These figures are considered similar or slightly higher compared to ours. In the latter article, the authors discussed that their estimate of the prevalence of proteinuria was likely to be an underestimate, which was attributed to participants who had a history of nephropathy at baseline and results recorded as unknown.

Proteinuria status, not always available in our study, has been known as a risk factor for mortality and cardiovascular diseases^{4,5}. Taken together with the reviewer's suggestions, we should describe the possibility of misclassification of proteinuria status as one of the limitations of this study.

Revisions of the manuscript:

Discussion (page 17):

'Although proteinuria has been known as a risk factor for mortality and cardiovascular diseases [28,29], we could not determine proteinuria status completely as reported previously [30,31]. Incomplete records on proteinuria may introduce a bias for proteinuria status and possibly influence the study results.'

References:

- 1) NHS Digital. National Diabetes Audit, 2015-2016. Report 1: Care Processes and Treatment Targets. 31 January 2017.
- 2) Dreyer G, Hull S, Aitken Z, Chesser A, Yaqoob MM. The effect of ethnicity on the prevalence of diabetes and associated chronic kidney disease. QJM. 2009;102(4):261-269.
- 3) Liang H, Kennedy C, Manne S, Lin JH, Dolin P. Monitoring for proteinuria in patients with type 2 diabetes mellitus. BMJ Open Diabetes Res Care. 2015;3(1):e000071.

- 4) Chronic Kidney Disease Prognosis Consortium., Matsushita K, van der Velde M, Astor BC, Woodward M, Levey AS, de Jong PE, et al. Association of estimated glomerular filtration rate and albuminuria with all-cause and cardiovascular mortality in general population cohorts: a collaborative meta-analysis. *Lancet*. 2010;375(9731):2073-2081.
- 5) Matsushita K, Coresh J, Sang Y, Chalmers J, Fox C, Guallar E, et al. Estimated glomerular filtration rate and albuminuria for prediction of cardiovascular outcomes: a collaborative meta-analysis of individual participant data. *Lancet Diabetes Endocrinol*. 2015;3(7):514-525.